# Land Use/Cover Change and Its Driving Mechanism in Thailand from 2000 to 2020

Yiming Wang [1,2], Yunfeng Hu [1,3,*], Xiaoyu Niu [1,2], Huimin Yan [1,3] and Lin Zhen [1,3]

1   State Key Laboratory of Resources and Environmental Information System, Institute of Geographic Sciences and Natural Resources Research, Chinese Academy of Sciences, Beijing 100101, China
2   School of Geosciences, Yangtze University, Wuhan 430100, China
3   College of Resources and Environment, University of Chinese Academy of Sciences, Beijing 100049, China
*   Correspondence: huyf@lreis.ac.cn

**Abstract:** Land use/cover change (LUCC) research is important for regional ecological conservation and sustainable development. There is a lack of exploration of long-time-series dynamics and driving mechanisms at the national scale in the study of land use/cover change in Thailand. Based on the Global Land Cover product with Fine Classification System in 30m (GLC_FCS30) dataset and socioeconomic statistics, we studied the temporal and spatial evolution characteristics and driving mechanisms of LUCC in Thailand from 2000 to 2020 through indicators and methods such as land use dynamic attitude, mapping of a Sankey diagram, principal component analysis, and multiple linear stepwise regression analysis. The results showed that: (1) Thailand has developed in terms of agriculture and forestry. In 2020, the cropland and forest areas accounted for 53.77% and 32.15% of the land area, respectively. (2) From 2000 to 2020, the area of rainfed cropland, irrigated cropland, and forest continued to shrink; the area of impervious surfaces expanded rapidly, and the area of shrubland, other cropland, and wetlands increased. (3) The LUCC process mainly occurred in the two-way conversion between forest and shrubland, rainfed cropland and irrigated farmland, forest and rainfed cropland, and forest and other farmland. The LUC with the largest area transformed into other land types was forest ($2.25 \times 10^4$ km$^2$), and the LUC with the largest area transformed from other land types transferring into the area was shrubland ($1.40 \times 10^4$ km$^2$). (4) From 2000 to 2020, the LUCC process in Thailand was mainly influenced by socio-economics and tourism. Gross population, main grain output, industrial value added, passenger income, and urban population were the key factors driving the LUCC in Thailand. Our research can provide the basis and decision support for the future planning and management of land in Thailand.

**Keywords:** LUCC; land mapping; spatial distribution; time-series analysis; correlation analysis; national development

## 1. Introduction

Land cover refers to the natural and biophysical properties of the earth's surface. Land use refers to the use status of the land or the social and economic attributes of the land. These two components constitute the dual properties of land [1]. With the advancement of global change research, it is gradually being recognized that land use/cover change (LUCC) is an important cause of global change [2,3]. LUCC changes the available energy, water availability, photosynthesis rates, nutrient levels, and surface roughness at the surface, with significant implications for human and ecosystem development [4].

In 1986, the ICSU (International Council of Scientific Unions) launched the IGBP (International Geosphere-Biosphere Programme) [5], which began the study of earth changes. In 1995, the IGBP and IHDP (International Human Dimensions Programme on Global Environmental Change) jointly proposed an LUCC research project on anthropogenic and global-change-driven impacts on land use/cover change and its environmental and

social impacts [6]. This made world land use and land cover changes a major theme in global environmental change research. In 2005, the IGBP and IHDP jointly launched the Global Land Project (GLP) with the core objective of measuring, modeling, and understanding coupled human–environment systems [7]. In 2014, the ICSU and ISSC (International Social Science Council) together with UNESCO (United Nations Educational, Scientific and Cultural Organization), UNEP (United Nations Environment Programme), and other international organizations, launched the "Future Earth" (2014–2023) to strengthen communication and cooperation between the natural and social sciences to address the challenges posed by global environmental change to regions, countries, and societies and to provide the necessary theoretical knowledge, research tools, and methods for global sustainable development [8].

Since the 21st century, with the development of remote sensing technology, geographic science, statistical methods, and software, more accurate data and advanced analysis methods have been provided for the study of land use/cover change. Gao et al. [9] used sampling analysis and correlation studies to analyze the spatial distribution and characteristics of LUCC in the United States and the effects of population and elevation on land use classes. Based on the analysis of land use/cover change, scholars have explored the driving mechanisms. Bai et al. [10] conducted an in-depth analysis of the holistic, hierarchical, and dynamic changes in LUCC driving forces and the dynamics of LULC under the action of driving forces. They preliminarily answered the questions on the dynamic sources of LUCC, the relationship between partial and combined forces within the driving force system, and the prevalent nonlinear feedback relationship between driving forces and land use change. It provided a new idea for the study of land use/land cover change dynamics at that time. Fernández et al. [11] studied the patterns and drivers of LUCC in the southern Ecuadorian Andes, illustrating the effects of land use policies, credit, and land tenure incentives, and demographics on LUCC. Arowolo et al. [12] analyzed land use/land cover change in Nigeria, performed a statistical modeling of the drivers of arable land change, and concluded that policy measures to increase agricultural productivity remain one of the best ways to reduce the pressure on Nigeria's increasingly scarce land resources and protect its natural ecosystems.

Thailand is an open country with frequent exchange with international countries. In 2017, Thailand's GDP ranked first among the five countries in the Indo-China Peninsula and eighth in Asia, with a strong economy. Walsh et al. [13] analyzed the LULC and NDVI (Normalized Difference Vegetation Index) changes in the Nang Rong region of northeastern Thailand at different spatial and temporal scales, indicating that social factors such as deforestation and agricultural extension had a strong influence. Wiriyanuwatkul et al. [14] quantified the magnitude of land use change in Thailand from 2000–2007. During this period, forest and cropland were lost, grassland, wetlands, and settlement area increased, forest land conversion occurred mainly in the northern region, the arable land loss occurred mainly in the central, eastern, and southern regions, and land conversion to settlements increased in all regions. Wijitkosum et al. [15] studied the spatial changes in land use in the area around the Huaxi Research Center in Thailand. Their study showed that the increase in forest area successfully reduced the severity of drought problems and desertification risk, providing an effective remedy for restoring degraded land and combating desertification. At present, studies on LUCC in Thailand are mainly regional and are usually conducted in provinces, watersheds, cities, and borders. There is a lack of spatial and temporal evolution analysis of LUCC in Thailand at a long-time-series and national scale. Moreover, there are a lack of quantitative statistical analysis and corresponding driving analysis models for the analysis of the driving mechanisms.

In order to analyze the spatio-temporal evolution of LUCC in Thailand on a long-term and national scale, quantitatively analyze its driving mechanism, and establish a corresponding regression equation, this paper took the whole area of Thailand as the research scope based on the long-term, high-resolution GLC_FCS30 (Global Land Cover product with Fine Classification System) data set, integrated key data such as climate and

national socio-economic data, and analyzed Thailand's land use/cover spatio-temporal evolution characteristics and dynamic changes. We synthesized key data such as the climate and national social economy and used a principal component analysis to obtain the main driving factors of Thailand's land use/cover change. We used a linear stepwise regression analysis to establish a quantitative regression equation to explore Thailand's land utilization/coverage change driving mechanisms. We attempted to answer the following three questions:

1. What are the spatial distribution patterns and temporal change characteristics of land use/cover in Thailand from 2000 to 2020?
2. What are the driving mechanisms of land use/cover change in Thailand from 2000 to 2020?
3. What are the uncertainties in the analysis of land use/cover change in Thailand?

## 2. Data and Methods

### 2.1. Study Area

Thailand (Figure 1) is located in the south-central part of the Indo-China Peninsula (5°30′–21° N, 97°30′–105°30′ E) bordered by the Gulf of Thailand in the southeast of the Pacific Ocean and the Andaman Sea in the southwest of the Indian Ocean. It borders Myanmar to the west and northwest, Laos to the northeast, Cambodia to the east, and Malaysia to the south. The total area of the country is $51.3 \times 10^4$ km$^2$ and the coastline is $0.27 \times 10^4$ km.

The overall topography of Thailand is high in the north and low in the south, sloping from northwest to southeast, with more than 50% being plains and lowlands. Topographically, it is divided into four natural regions: the northern mountainous region, the central Mekong plain, the northeastern Korat plateau, and the southern Malay Peninsula. Most of Thailand has a tropical monsoon climate, with the year divided into three seasons: the hot season (February to mid-May), the rainy season (June to mid-October), and the cool season (November to next February). The average annual temperature is about 27.7 °C, and the maximum temperature can be over 40 °C. The average annual precipitation is 1100 mm, and the average humidity is 66–82%.

Thailand has 77 provinces. Among them, Bangkok, the capital city, is a municipality located on the banks of the Chao Phraya River and is the largest city in Thailand and the second largest city in southeast Asia, as well as the political, economic, cultural, and transportation center of Thailand. The National Statistical Office of Thailand (http://www.nso.go.th/, accessed on 30 September 2022) has divided the country into 7 major regions, namely the central region (CR), southern region (SR), western region (WR), eastern region (ER), northern region (NR), northeastern region (NER), and Bangkok and its vicinity (BV), based on the topographical features of the country and the economic development of the country [16].

Thailand is generally a service country. In 2020, the ratio of the added value of the primary, secondary, and tertiary industries to the national GDP was 8.69%, 30.51%, and 60.69%, respectively. Although the share of agriculture has gradually declined in recent years, it still occupies an important position in Thai industry. Thailand has a vast arable land area of about $15 \times 10^4$ km$^2$, accounting for 31% of the total land area. Thailand has rich fishery resources and vast sea areas. The Gulf of Thailand and the Andaman Sea are blessed with natural marine fishing grounds. Bangkok, Songkhala, and Phuket are important fishing centers and distribution points for fishery products. Thailand's agricultural products are one of the major sources of foreign exchange earnings. Major agricultural exports include rice, natural rubber, cassava, corn, sugar cane, and tropical fruits (including bananas, pineapples, durian, etc.). Thailand is the world's largest exporter of several agricultural products such as natural rubber, durian, and mangosteen. Thailand's industry is export-oriented. It has a complete automobile industry chain and is the third largest exporter of automobiles in Asia after Japan and South Korea. Tourism is the mainstay of Thailand's service industry. The country is rich in tourism resources, with

more than 500 well-known attractions; the main tourist locations are Bangkok, Phuket, Phattaya, Chiang Mai, Chiang Rai, and Hua Hin.

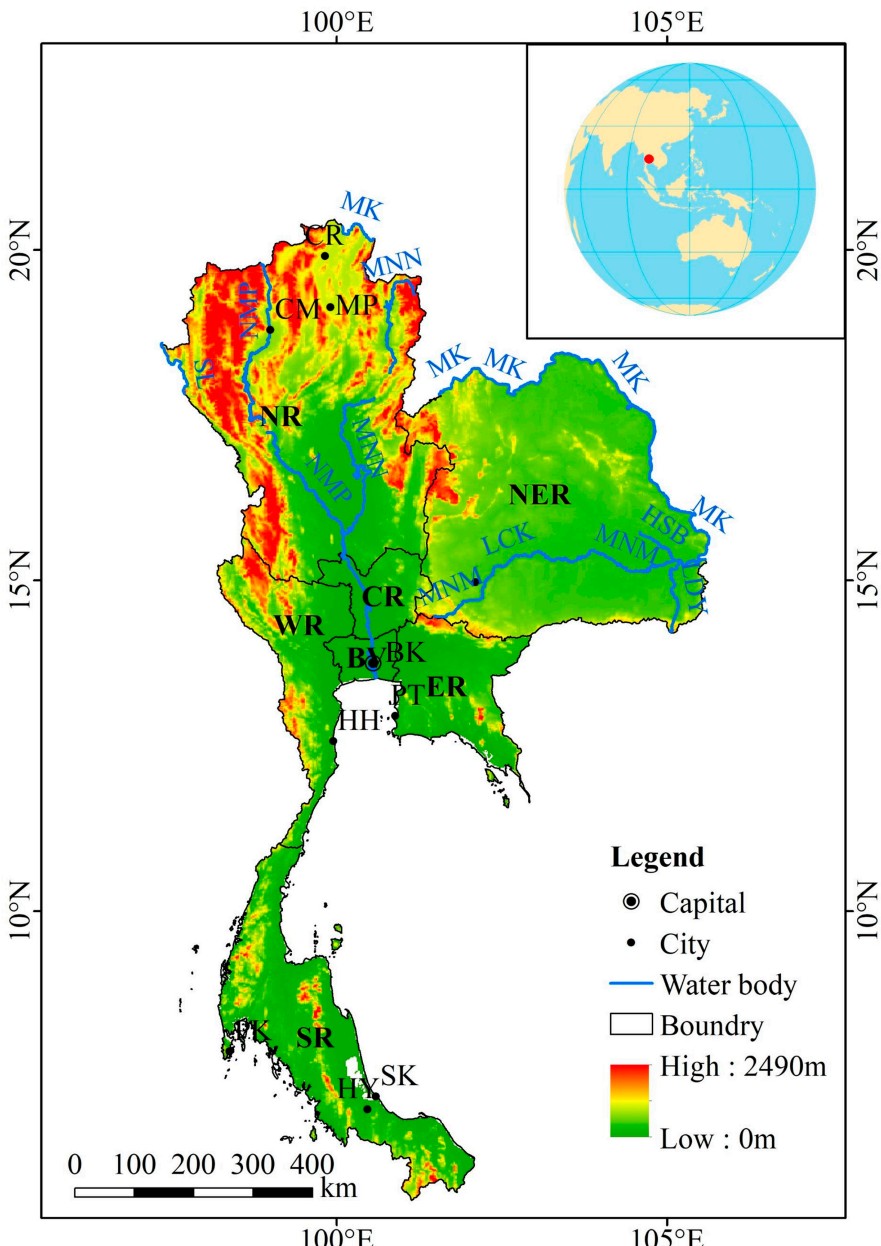

**Figure 1.** Location and topography map of Thailand. Region: NR: northern region; NER: northeastern region; CR: central region; WR: western region; ER: eastern region; BV: Bangkok and its vicinity; SR: southern region. Capital and city: BK: Bangkok; CM: Chiang Mai; CR: Chiang Rai; HH: Hua Hin; HY: Hat Yai; MP: Muang Phayao; NP: Nakhon Potchasima; PK: Phuket; PT: Phattaya; SK: Songkhala. River: HSB: Huai Se Bok; LCK: Lam Chiang Krai; LDY: Lam Dom Yai; MK: Mekong; MNM: Mae Nam Mun; MNN: Mae Nam Nan; NMP: Nam Mae Ping; SL: Salween.

### 2.2. Data Sources

The LULC dataset (GLC_FCS30) used in the study was derived from the China Earth Big Data Science Project Data Sharing Service System (https://data.casearth.cn/sdo/list?searchKey=GLC_FCS30, accessed on 29 September 2022). The dataset covers the period 1985–2020 and is updated every 5 years for a total of 8 periods, with a spatial resolution of 30 m [17]. Among them, there are 9 primary surface cover types and 30 secondary fine land

cover types, and the overall accuracy of the 8 primary land types and the 30 secondary land types reaches 82.5% and 68.7%, respectively [18].

Since Thailand is located in the tropics and subtropics, the water and heat conditions are very good, and the vegetation coverage is extremely rich. In the land use/land cover classification system of GLC_FCS (Table 1), there are 20 secondary land types in Thailand. The land use/land cover classification system in Thailand (Table 2) was reclassified on the basis of the land use/land cover classification system of GLC_FCS. However, there are many LUC types that do not exist in Thailand (such as ice and snow, permafrost, etc.). At the same time, since grassland, bare areas, and shrubland are too small, they are suitable to be combined into one land type, while some types of LUC with a large area and important status (such as cropland) needed to be further subdivided. However, some types of large and important LUC (such as cropland) need to be further subdivided (level 2) for analysis. Therefore, after comprehensively considering these factors, we obtained the land use/land cover classification system in Thailand (Table 2). In our follow-up studies, statistics and analysis were performed using the land use/land cover classification system in Thailand (Table 2).

**Table 1.** Land use/land cover classification system of GLC_FCS.

| Code | Level 1 Classes | LC ID | Level 2 Classes |
|---|---|---|---|
| 1 | | 10 | Rainfed cropland |
| | Cropland | 11 | Herbaceous cover |
| | | 12 | Tree or shrub cover(orchard) |
| | | 20 | Irrigated cropland |
| 2 | | 51 | Open evergreen broadleaved forest |
| | | 52 | Closed evergreen broadleaved forest |
| | | 61 | Open deciduous broadleaved forest (0.15 < fc < 0.4) |
| | | 62 | Closed deciduous broadleaved forest (fc > 0.4) |
| | Forest | 71 | Open evergreen needle-leaved forest (0.15 < fc < 0.4) |
| | | 72 | Closed evergreen needle-leaved forest (fc > 0.4) |
| | | 81 | Open deciduous needle-leaved forest (0.15 < fc < 0.4) |
| | | 82 | Closed deciduous needle-leaved forest (fc > 0.4) |
| | | 91 | Open mixed leaf forest (broadleaved and needle-leaved) |
| | | 92 | Closed mixed leaf forest (broadleaved and needle-leaved) |
| 3 | | 120 | Shrubland |
| | Shrubland | 121 | Evergreen shrubland |
| | | 122 | deciduous shrubland |
| 4 | Grassland | 130 | Grassland |
| 5 | Wetlands | 180 | Wetlands |
| 6 | Impervious surfaces | 190 | Impervious surfaces |
| 7 | | 140 | Lichens and mosses |
| | | 150 | Sparse vegetation (fc < 0.15) |
| | | 152 | Sparse shrubland (fc < 0.15) |
| | Bare areas | 153 | Sparse herbaceous (fc < 0.15) |
| | | 200 | Bare areas |
| | | 201 | Consolidated bare areas |
| | | 202 | Unconsolidated bare areas |
| 8 | Water body | 210 | Water body |
| 9 | Permanent ice and snow | 220 | Permanent ice and snow |
| | | 250 | Filled value |

**Table 2.** Land use/land cover classification system in Thailand.

| Code | Level 1 Classes | LC ID | Level 2 Classes |
|---|---|---|---|
| 1 | Rainfed cropland | 10 | Rainfed cropland |
| 2 | Irrigated cropland | 20 | Irrigated cropland |
| 3 | Other cropland | 11 | Herbaceous cover |
|  |  | 12 | Tree or shrub cover (orchard) |
| 4 | Forest | 51 | Open evergreen broadleaved forest |
|  |  | 52 | Closed evergreen broadleaved forest |
|  |  | 61 | Open deciduous broadleaved forest (0.15 < fc < 0.4) |
|  |  | 62 | Closed deciduous broadleaved forest (fc > 0.4) |
|  |  | 71 | Open evergreen needle-leaved forest (0.15 < fc < 0.4) |
|  |  | 72 | Closed evergreen needle-leaved forest (fc > 0.4) |
| 5 | Shrubland | 120 | Shrubland |
|  |  | 121 | Evergreen shrubland |
|  |  | 122 | deciduous shrubland |
|  |  | 130 | Grassland |
|  |  | 150 | Sparse vegetation (fc < 0.15) |
|  |  | 200 | Bare areas |
| 6 | Wetlands | 180 | Wetlands |
| 7 | Impervious surfaces | 190 | Impervious surfaces |
| 8 | Water body | 210 | Water body |
|  |  | 220 | Permanent ice and snow |

In Thailand from 2000 to 2020, the average annual temperature (X1) was obtained from the GLDAS-2.1 dataset [19], and the total annual precipitation (X2) was obtained from the PERSIANN-CDR dataset [20] with a spatial resolution of 0.25 radians. The total annual precipitation and the average annual temperature of the study area were obtained by calculating the day-by-day data in the dataset. The quality of these two data sets has been recognized by scholars and has been applied to many scientific studies. For example, the GLDAS data set was used in the research of Ni et al. [21] and Jiang et al. [22]. The PERSIANN-CDR data set is used in the research of Bathelemy et al. [23] and Yang et al. [24].

Statistics on social and economic development (X3–X14) (2000–2020) were obtained from the World Bank (https://data.worldbank.org.cn/country/thailand?view=chart, accessed on 29 September 2022), and the remaining data (X15–X17) were obtained from the World Food and Agriculture Organization (https://www.fao.org/faostat/en/#country/216, accessed on 29 September 2022). Of these, travel income excluded the cost of international transportation of travelers, passenger income excluded passenger services provided by resident carriers to non-residents within the economy, and international travel income included the cost of international transportation paid to domestic carriers and any other advance payments made to obtain goods or services in the destination (country) of travel. The export of agricultural raw materials did not include fuel, crude fertilizer, except coal, petroleum, precious stones, ore containing metals, and metal scrap. The main grain output included the output of rice, cassava, and maize, with rice output accounting for about 50%. The main fruit output included the output of oil palm fruits, sugar cane, bananas, and pineapples, with sugar cane production being much larger than that of other fruits, accounting for more than 80%.

According to the above statistical data, combined with the law of general social and economic development and the actual development of Thailand (Section 2.1 Study Area), this paper established the driving factor system of land use/cover change (Table 3).

**Table 3.** Indicators and their categories of economic and social development statistics from 2000 to 2020.

| Category | Index | Unit |
|---|---|---|
| Climate | X1 Average annual temperature | °C |
| | X2 Total annual precipitation | mm |
| Social development | X3 Gross population | Ten thousand |
| | X4 Rural population | Ten thousand |
| | X5 Urban population | Ten thousand |
| | X6 Urbanization rate | % |
| Economic development | X7 Gross domestic product (GDP) | USD billion (current USD) |
| | X8 Agricultural value added | USD billion (current USD) |
| | X9 Industrial value added | USD billion (current USD) |
| | X10 Export of agricultural raw materials | USD billion (current USD) |
| | X11 Total fisheries production | Ten thousand t |
| | X12 International tourism income | USD billion (current USD) |
| | X13 Passenger income | USD billion (current USD) |
| | X14 Travel income | USD billion (current USD) |
| | X15 Main grain output | Ten thousand t |
| | X16 Natural rubber output | Ten thousand t |
| | X17 Main fruit output | Ten thousand t |

Note: X1–X3 were from the GLDAS-2.1 and PERSIANN-CDR datasets; X4–X14 were from the World Bank; and X15–X17 were from the World Food and Agriculture Organization.

Generally speaking, the driving factors of LUC changes in a country and region are climatic, social, and economic factors. Based on previous studies [11,12,25] in this regard, we selected climatic factors such as X1–X2 and social and economic development factors such as X3–X9. At the same time, according to the actual situation in Thailand, we made some targeted supplements and adjustments in terms of the key driving factors. As a result, characteristic factors such as X10–X17 were formed.

For climate change, researchers generally consider studying from two aspects: temperature and precipitation. For Thailand, this may also include some extreme climates (such as extreme heat, hurricanes, etc.). However, on a 20-year scale such as that involved in this study, changes in extreme climate may not be fully reflected. It is generally believed that a 20-year time scale can only reflect fluctuations in extreme climate. For social development, it is mainly reflected in the development of the population, which mainly included X3–X6 in the indicators. Macroeconomic development indicators included X7–X17 in the indicators. Among them, the most typical economic development factor was X7–X9, while X10–X17 was more focused on industrial development. At the same time, the eight indicators such as X10–X17 were carefully selected after considering the characteristics of Thailand's developed tourism industry and distinctive agricultural industries (such as fruit exports and natural rubber exports).

### 2.3. LUCC Analysis Method

We analyzed the land use/cover changes using integrated land use dynamics, single land use dynamics, and the land use transfer matrix.

The integrated land use dynamic degree (*LC*) was used to analyze the overall quantitative changes in land use types in the study area [26], characterizing the rate of regional land use change. The calculation method is as follows:

$$LC = \left\{ \sum_{i=1}^{n} \left( \frac{\Delta U_i}{U_i} \right) \right\} \times \frac{1}{T} \times 100\% \tag{1}$$

where *LC* is the integrated land use dynamic degree during the study period; $\Delta U_i$ is the area of change of type *i* land use/cover type at the beginning and the end of the study;

$U_i$ is the area of type $i$ land use/cover type at the beginning of the study; T is the study period; and $n$ is the number of land use/cover types.

The single land use dynamic degree ($K$) can quantitatively describe the change in a certain land use type within a certain time frame in a region. It is important for comparing regional differences in land use change and predicting future land use change trends [27]. The calculation method is as follows:

$$K = \frac{U_b - U_a}{U_a} \times \frac{1}{T} \times 100\%$$

(2)

where $K$ is the single dynamic degree of a certain land use/cover type during the study period; $U_a$ and $U_b$ denote the area of a certain land type at the beginning and end of the study, respectively; and T is the study period.

The land use transfer matrix can show the structural characteristics of regional land use change in a comprehensive and concrete way, reflecting the direction of land use change guided by human activities. This method reflects the process of state transformation of a sub-stable system from T moments to T+1 moments under a certain time interval, thus it can better reveal the spatio-temporal evolution of land use patterns [28]. Its expression is:

$$S_{ij} = \begin{bmatrix} S_{11} & S_{12} & \cdots & S_{1n} \\ S_{21} & S_{22} & \cdots & S_{2n} \\ \vdots & \vdots & \vdots & \vdots \\ S_{n1} & S_{n2} & \cdots & S_{nn} \end{bmatrix}$$

(3)

where S denotes the land area; $i$ and $j$ denote the land use/cover types at the beginning and end of the study period, respectively; and $n$ is the number of land use/cover types.

### 2.4. Driving Mechanism Analysis Method

When the driving mechanism of LUCC is analyzed, there will be many driving factors. Although multiple drivers make the analysis results more comprehensive, they can create data redundancy, making the analysis process more complex and computationally intensive. In addition, there may be a correlation between the data, making the analysis results of the driving mechanism less reliable.

Principal components analysis (PCA) uses the idea of dimensionality reduction to transform multiple metrics into a few composite metrics while keeping the information of the data as much as possible without loss and eliminating data redundancy [29]. It transforms a given set of correlated variables into another set of uncorrelated variables by a linear transformation. These new variables are arranged in order of decreasing variance. The total variance of the variables is kept constant in the mathematical transformation. The first variable has the largest variance and is called the first principal component ($F_1$). The second variable with the second largest variance that is uncorrelated with the first variable is called the second principal component ($F_2$). By analogy, there are $n$ principal components for $n$ variables. After analysis by principal component analysis, we can rename the extracted principal component factors to obtain reasonable explanatory variables based on expertise and the unique meanings reflected by the indicators [30]. Each principal factor is selected as follows:

$$\begin{cases} F_1 = c_{11}Z_1 + c_{12}Z_2 + \cdots + c_{1n}Z_n \\ F_2 = c_{21}Z_1 + c_{22}Z_2 + \cdots + c_{2n}Z_n \\ \qquad\qquad \vdots \\ F_n = c_{n1}Z_1 + c_{n2}Z_2 + \cdots + c_{nn}Z_n \end{cases}$$

(4)

where: $F_i$ denotes $i$ principal component, $i = 1, 2, \ldots, n$; $c$ is the eigenvector corresponding to the eigenvalues of the covariance array; $Z$ is the normalized value of the original variables; and for each $i$ there is $c_{i1}^2 + c_{i2}^2 + \cdots + c_{in}^2 = 1$.

Land use/cover change is influenced by a variety of factors such as climate, social, and economic development. The basic idea of multiple linear stepwise regression analysis is that variables go in and out. The specific method is to introduce variables one by one. After each variable is introduced, the selected variables are tested one by one. When the originally introduced variable becomes insignificant due to the later introduced variable, it will be eliminated to ensure that only each significant variable is included in the regression equation before new variables are introduced, and this method is repeated until no significant independent variables are introduced into the regression equation, and no non-significant independent variables are removed from the regression equation. It is used to build optimal or appropriate regression models to study the dependencies between variables in greater depth [31]. To improve the prediction accuracy of the dependent variable, we used multiple linear stepwise regression analysis to analyze multiple influencing factors. The multiple regression model is as follows.

$$Y = \beta + \alpha_1 \, X_1 + \alpha_2 \, X_2 + \ldots + \alpha_n X_n \tag{5}$$

where: $\alpha_1, \alpha_2, \ldots, \alpha_n$ denote the correlation coefficients; $\beta$ is the constant term; $Y$ is the dependent variable; and $X_1 - X_n$ is the independent variable.

## 3. Results

### 3.1. Spatial Distribution

From the statistics of the various types of land area in Thailand in 2020 (Figure 2), it can be seen that agricultural land in Thailand was the most widely distributed ($29.81 \times 10^4$ km$^2$, 53.77%), which was more than 1/2 of the total land area of the country. Most of them were rainfed cropland ($18.01 \times 10^4$ km$^2$, 32.48%), and a few were other cropland ($6.51 \times 10^4$ km$^2$, 11.75%) and irrigated cropland ($5.29 \times 10^4$ km$^2$, 9.54%). Forest ($17.82 \times 10^4$ km$^2$, 32.15%) was the second largest land cover type in Thailand and was widely distributed, exceeding 1/3 of the total land area of the country. The third largest land cover type was shrubland ($4.88 \times 10^4$ km$^2$, 8.80%). The total area of impervious surface, water bodies, and wetlands ($2.92 \times 10^4$ km$^2$) accounted for only 5.27% of the national land area, of which impervious surface was $1.72 \times 10^4$ km$^2$ (3.10%), water bodies was $0.84 \times 10^4$ km$^2$ (1.52%), and wetlands was $0.36 \times 10^4$ km$^2$ (0.65%).

Spatially, rainfed cropland was mainly distributed in the northeastern Korat Plateau and the central Mae Nam River plain, with a small amount in the northern mountains. Irrigated cropland was mainly located in Bangkok and vicinity, central plain areas, and the diversions of the Mae Nam Na and Nam Mae Ping rivers in the northern region, with scattered distribution in Chiang Rai in the northern region and Phattaya in the eastern region. Other cropland (herbaceous cover and orchard) was mainly located in the areas of Songkhala, Hat Yai, etc., in the southern Malay Peninsula. Forest was mainly located in the northern mountainous areas of Chiang Mai and Muang Phayao, with some in the western border areas with Myanmar and the southern Malay Peninsula and eastern coastal areas. Shrubland was scattered mainly in the northern and western mountains and sporadically along the Mekong coast. Impervious surfaces showed a blocky distribution concentrated in large cities. It was mostly distributed in Bangkok and its vicinity, followed by the northern mountainous Chiang Mai area, and then, to a lesser extent, in the cities of Chiang Rai and Muang Phayao in the northern region, Nakhon Potchasima in the northeastern region, and Phattaya in the eastern region. Water bodies were mainly distributed in the western and northern mountainous areas and to a lesser extent in the eastern Gulf of Thailand coast. The distribution of wetlands was very small, showing a dotted distribution mainly in the southwest Indian Ocean along the Andaman Sea and the southeast Gulf of Thailand coastal areas.

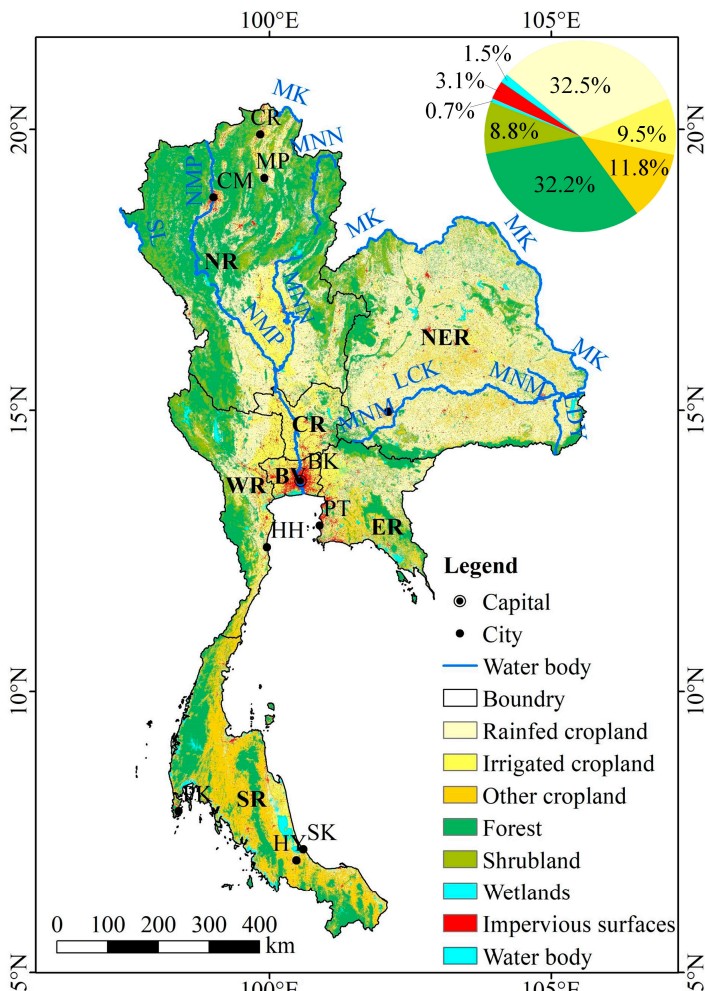

**Figure 2.** Land use/cover map of Thailand in 2020. Region: NR: northern region; NER: northeastern region; CR: central region; WR: western region; ER: eastern region; BV: Bangkok and its vicinity; SR: southern region. Capital and city: BK: Bangkok; CM: Chiang Mai; CR: Chiang Rai; HH: Hua Hin; HY: Hat Yai; MP: Muang Phayao; NP: Nakhon Potchasima; PK: Phuket; PT: Phattaya; SK: Songkhala. River: HSB: Huai Se Bok; LCK: Lam Chiang Krai; LDY: Lam Dom Yai; MK: Mekong; MNM: Mae Nam Mun; MNN: Mae Nam Nan; NMP: Nam Mae Ping; SL: Salween.

### 3.2. Dynamic Changes

Shrubland expanded the most across Thailand from 2000 to 2020 ($0.72 \times 10^4$ km², 0.87%) followed by impervious surfaces, which also showed an increasing trend ($0.58 \times 10^4$ km², 2.56%), and wetlands increased the least ($0.03 \times 10^4$ km², 0.48%). Cropland as a whole showed a decrease ($-0.45 \times 10^4$ km², $-0.92\%$). Among them, both rainfed cropland and irrigated cropland decreased (the decreases were $-0.40 \times 10^4$ km², $-0.11\%$ and $-0.03 \times 10^4$ km², $-0.03\%$), and other farmland (herbaceous cover and orchard) showed an increase ($+0.16 \times 10^4$ km², 0.12%). The largest absolute area reduction was forest ($-0.72 \times 10^4$ km², $-0.16\%$). Water bodies showed a decreasing trend ($-0.02 \times 10^4$ km², $-0.11\%$).

From 2000 to 2020, the spatial distribution pattern of integrated land use dynamic degree in Thailand showed an increasing trend centered on the central Chao Phraya River plain in all directions (Figure 3). There were provinces with higher dynamic degrees distributed in the northern mountains, northeastern highlands, eastern coastal areas, and southern Malay Peninsula of Thailand. Among the seventy-seven provinces, five provinces had an integrated land use dynamic degree of more than 100%, namely Surat Thani (southern region), Trat (southeastern region), Udon Thani (eastern region), Tak (western

region), and Surin (eastern region). Their combined land use dynamics were as high as 3067.43%, 722.88%, 325.19%, 280.93%, and 269.71%, respectively, mainly due to the increase in shrubland. Only Chachoengsao (33.54%) had an integrated land use dynamic degree of 33% or more, mainly due to the increase in other cropland, wetlands, and impervious surfaces. The provinces with an integrated land use dynamic degree between 15–33% were mainly located around Bangkok, such as Samut Prakan (24.05%, mainly caused by an increase in rainfed cropland, wetlands, and impervious surfaces and a decrease in forest), Rayong (20.54%, mainly caused by an increase in shrubland, wetlands, and impervious surfaces), Ang Thong (19.66%, mainly caused by an increase in wetlands and irrigated cropland), and Kanchanaburi (24.33%, mainly caused by an increase in other cropland, shrubland, and water bodies and a decrease in wetlands). ThisI wass also located in the southern coastal areas such as Prachuap Khiri Khan (15.37%, mainly caused by an increase in impervious surfaces and a decrease in forest) and Changwat Phuket (31.95%, mainly caused by an increase in other cropland and wetlands). The rest were distributed in Chiang Mai in the northeastern mountains (25.00%, mainly caused by an increase in shrubland, wetlands, and other cropland) and Buriram on the border with Cambodia (18.85%, mainly caused by an increase in other cropland and wetlands). The rest of the provinces had an integrated land use dynamic degree of less than 15% and a relatively inactive land use/cover change.

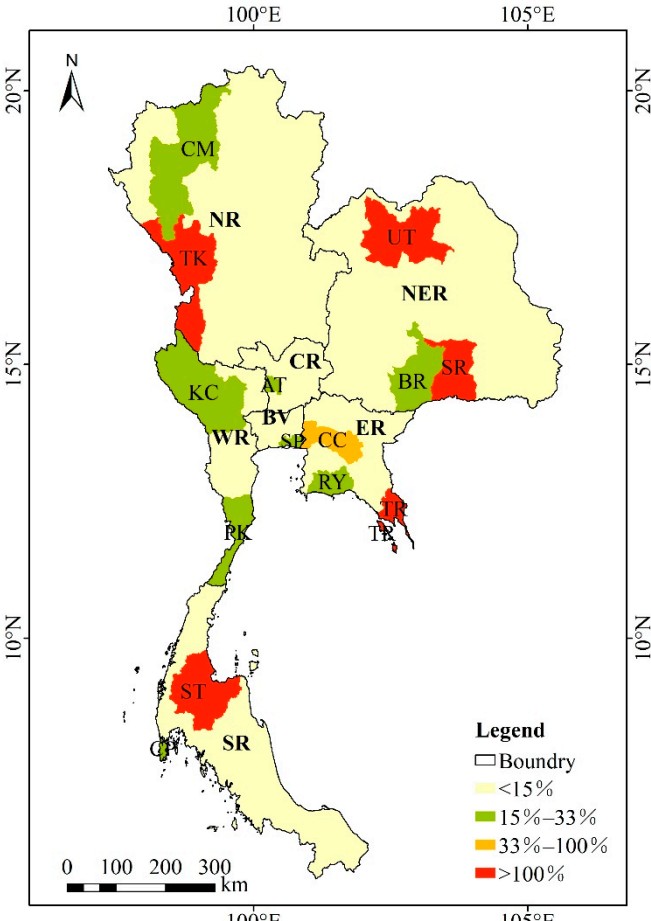

**Figure 3.** The spatial distribution map of the integrated dynamic degree of LULC in Thailand from 2000 to 2020. Region: NR: northern region; NER: northeastern region; CR: central region; WR: western region; ER: eastern region; BV: Bangkok and its vicinity; SR: southern region. Province: AT: Ang Thong; BR: Buriram; CC: Chachoengsao; CP: Changwat Phuket; CM: Chiang Mai; KC: Kanchanaburi; PK: Prachuap Khiri Khan; RY: Rayong; SP: Samut Prakan; ST: Surat Thani; SR: Surin; TK: Tak; TR: Trat; UT: Udon Thani.

### 3.3. Source and Destination

An analysis of the area and composition of land use/cover change in Thailand from 2000–2020 (Figure 4) showed that from 2000 to 2010, the total area of land use/cover type transformation (transfer in or out) was $4.99 \times 10^4$ km$^2$, and from 2010 to 2020, the total area of land use/cover type transformation (transfer in or out) was $5.16 \times 10^4$ km$^2$. The main land use/cover change processes that occurred during these two time periods were the interconversion between forest and shrubland, rainfed cropland and irrigated cropland, forest and rainfed cropland, and forest and other cropland.

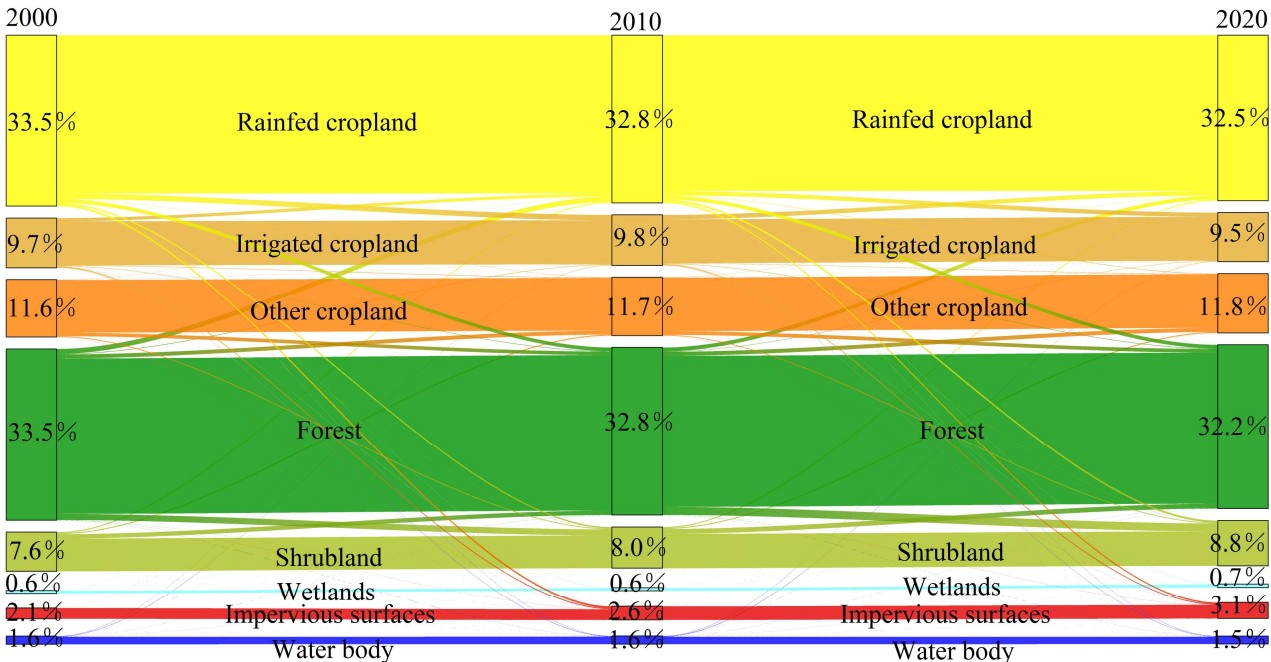

**Figure 4.** Land use/cover transfer plot of Thailand during 2000–2020.

From 2000 to 2020, the total area of rainfed cropland (Figure 5A) transferred out was $1.95 \times 10^4$ km$^2$, mainly being transferred out to irrigated cropland (36.28%, $0.71 \times 10^4$ km$^2$) and forest (26.30%, $0.51 \times 10^4$ km$^2$). It was distributed throughout the country, mainly in the northern mountains, Bangkok and its vicinity, and the western and southern regions, with a small scattered distribution in the southeastern region. The total area of irrigated cropland transferred out was $0.91 \times 10^4$ km$^2$, mainly being transferred out as rainfed cropland (52.90%, $0.48 \times 10^4$ km$^2$) and impervious surfaces (21.77%, $0.20 \times 10^4$ km$^2$) and was mainly concentrated in the south of Bangkok and its vicinity. The total area transferred from other cropland was $0.78 \times 10^4$ km$^2$, mainly from forest (62.04%, $0.48 \times 10^4$ km$^2$) and shrubland (16.12%, $0.13 \times 10^4$ km$^2$), and it was mainly concentrated in Chumphon and Rayong in the southern region.

From 2000 to 2020, the forest (Figure 5B) was mainly transferred out to shrubland (45.17%, $1.02 \times 10^4$ km$^2$), rainfed cropland (29.33%, $0.66 \times 10^4$ km$^2$), and the other cropland (21.48%, $0.48 \times 10^4$ km$^2$). This land use transition was mainly distributed in the northern and western mountainous areas and southern coastal areas. Shrubland (Figure 5C) was mainly transferred from forest (72.89%, $1.02 \times 10^4$ km$^2$) and rainfed cropland (18.97%, $0.27 \times 10^4$ km$^2$). This land use transition was mainly distributed in the northern and western mountainous areas. Wetlands (Figure 5D) were mainly transferred from rainfed cropland (32.05%, $0.02 \times 10^4$ km$^2$) and water bodies (15.76%, $0.01 \times 10^4$ km$^2$). Impervious surfaces (Figure 5E) were mainly transferred from rainfed cropland (56.81%, $0.33 \times 10^4$ km$^2$) and irrigated cropland (34.55%, $0.20 \times 10^4$ km$^2$). This land use transition was mainly distributed in Bangkok and its vicinity. The water bodies (Figure 5F) were

mainly transformed from irrigated cropland (42.99%, $0.04 \times 10^4$ km$^2$) and rainfed cropland (37.15%, $0.03 \times 10^4$ km$^2$).

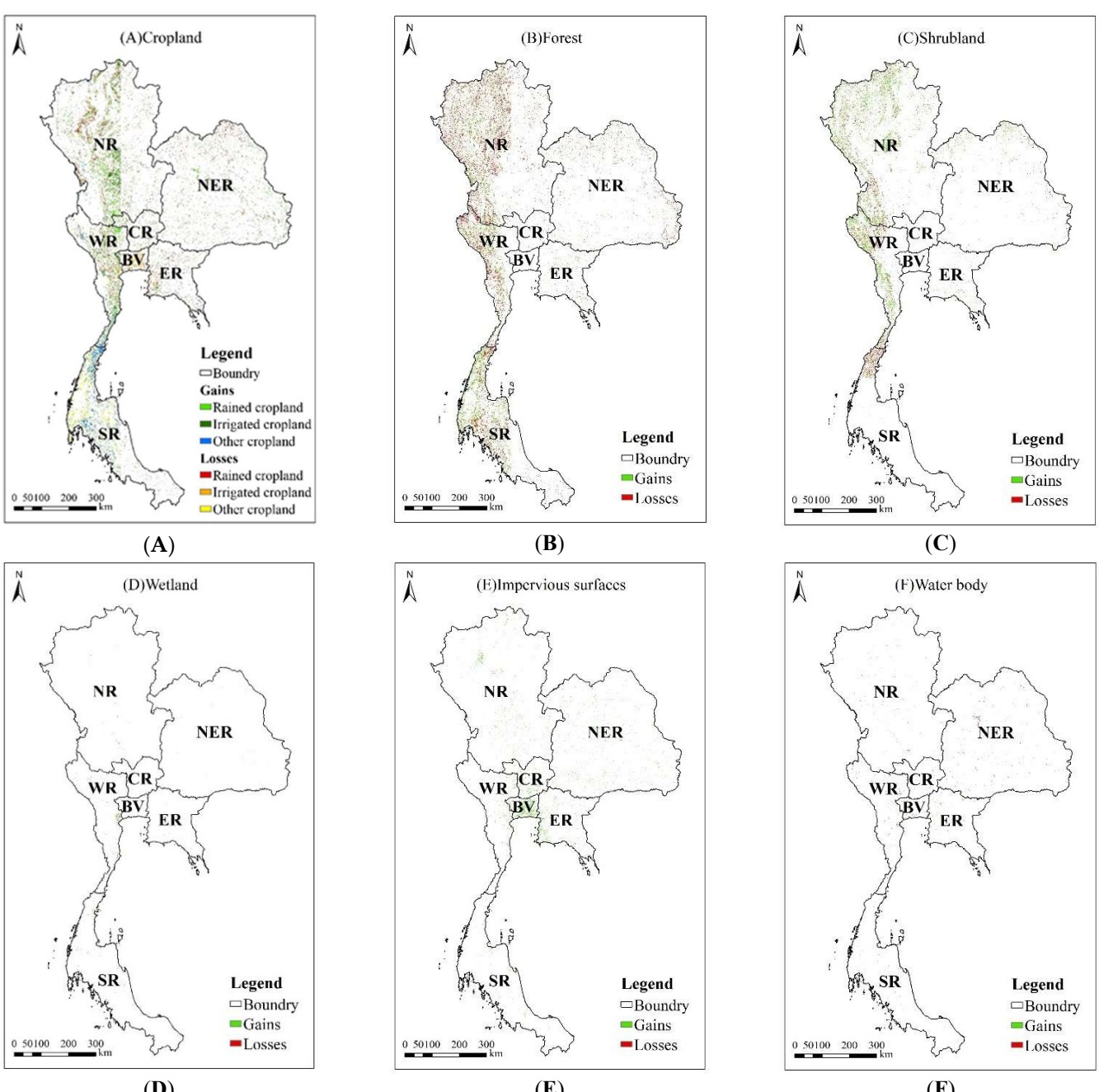

**Figure 5.** Spatial distributions of different LULC gains/losses during 2000–2020. NR: northern region. (**A**) Cropland; (**B**) Forest; (**C**) Shrubland; (**D**) Wetland; (**E**) Impervious surfaces; (**F**) Water body. Region: NER: northeastern region; CR: central region; WR: western region; ER: eastern region; BV: Bangkok and its vicinity; SR: southern region.

### 3.4. Climate Change, Social, and Economic Development Trajectories

From 2000 to 2020, Thailand's climate in general showed a slight warming and drying trend (Figure 6A). The average temperature in Thailand was 24.23 °C with a general upward trend (1.5 °C/10a), increasing especially significantly after 2011. The average total annual precipitation was 1733.45 mm, showing a general fluctuating downward trend (−9.3 mm/a). The total precipitation reached a peak in 2011 (2063.40 mm). In 2000, the

total precipitation reached 1956.88 mm. The rest of the years fluctuated within the range of 1500–1800 mm.

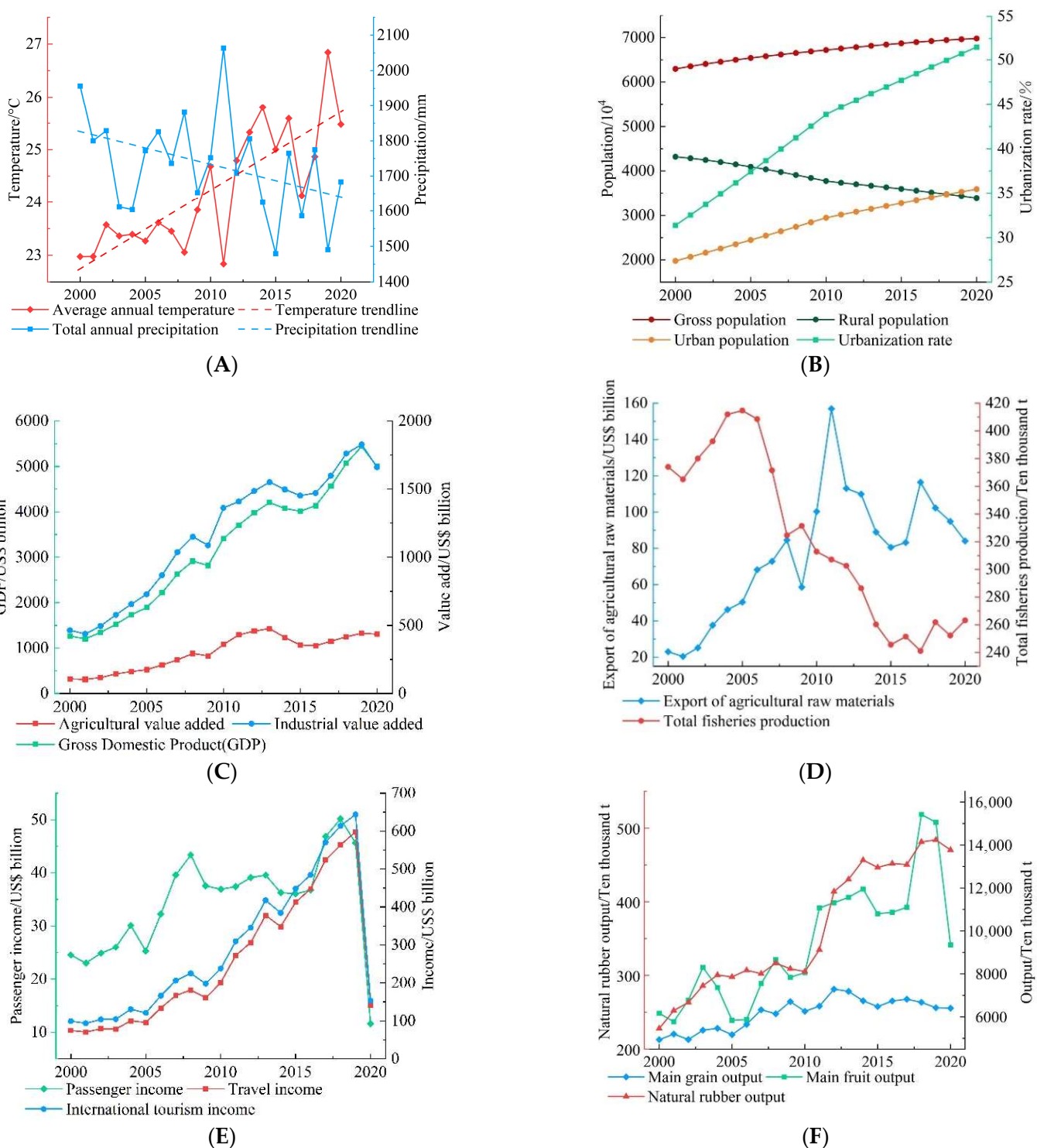

**Figure 6.** Development status of Thailand from 2000 to 2020. (**A**) Climatic factors; (**B**) population factor; (**C**) economic factor; (**D**) production and export factor; (**E**) tourism factor; and (**F**) agriculture factor.

From 2000–2020, Thailand's gross population and urban population continued to grow (Figure 6B). The gross population of the country grew from 62,952,600 in 2000 to 66,942,700 in 2020, an increase of 6,847,300, with an annual growth rate of 0.52%. The urban

population grew from 19,758,300 in 2000 to 35,898,100 in 2020, an increase of 16,139,800, with an annual growth rate of 3.03%. The urbanization rate kept accelerating, increasing from 31.39% in 2000 to 51.43% in 2020, but the growth rate slowed down after 2010. The rural population continued to decrease, from 43,194,300 in 2000 to 33,919,900 in 2020, a decrease of 9,292,500.

From 2000–2020, Thailand's overall rapid economic growth (Figure 6C). The national gross domestic product (GDP) increased from USD 1263.92 billion in 2000 to USD 4996.82 billion in 2020, with an annual growth rate of 7.11%. Agriculture developed more slowly, and industry developed rapidly. The agricultural value added increased from USD 107.43 billion in 2000 to USD 434.61 billion in 2020, with an average annual growth of USD 18.73 billion. The industrial value added increased from USD 463.47 billion in 2000 to USD 1661.09 billion in 2020, with an annual growth rate of 6.59%. However, the GDP, agricultural value added, and industrial value added all declined slightly in 2020, indicating that the global new COVID-19 epidemic in 2020 had an impact on Thailand's economy.

From 2000 to 2020, Thailand's total fishery production showed a general trend of rising, then falling, and then slowly fluctuating upward (Figure 6D). It peaked in 2005 (4,147,200 tons), then declined year by year, and fluctuated after 2015. The export of agricultural raw materials changed significantly in value, showing an increasing trend between 2000 and 2008. It declined in 2009, followed by a sudden increase in the following 2 years. It peaked in 2011 (USD 156.96 billion). Except for a sudden increase to USD 116.43 billion in 2017, it showed a decreasing trend in the other years.

From 2000–2019, Thailand's international tourism development showed an overall upward trend. However, international tourism was hit hard in 2020 by the global new COVID-19 epidemic (Figure 6E). International tourism income and travel income both peaked in 2019 (USD 643.71 billion and USD 598.10 billion). In addition, they differed by a small value each year, and the trend was also the same. Passenger income peaked in 2018 (USD 50.17 billion) and grew faster between 2005–2008 and 2016–2018.

There was an overall upward trend in Thailand's output of main grain, fruit, and natural rubber from 2000–2020 (Figure 6F). The main grain output grew slowly, reaching a peak in 2012 (72,896,200 tons), and it then decreased slowly thereafter. The main fruit output showed a fluctuating upward trend, peaking in 2018 (154,225,100 tons) and dropping abruptly to 93,517,900 tons in 2020 due to the global new COVID-19 epidemic. Natural rubber output showed a significant increase, peaking in 2019 (4,840,000 tons).

### 3.5. Driving Mechanism

Since the magnitudes of each driving factor shown in Table 3 were not the same, we standardized these indicator data using the standard deviation standardization method before conducting the principal component analysis.

From the score matrix of the principal component analysis (Table 4), it can be seen that the economic, social, and climate driving factors of Thailand could be divided into two dimensions from 2000 to 2020. F1 overall could be categorized as a socioeconomic dimension, which showed a very strong positive correlation with eight driving factors, including the average annual temperature (X1), gross population (X3), urban population (X5), urbanization rate (X6), GDP (X7), agricultural value added (X8), industrial value added (X9), and main grain production (X15), and a very strong negative correlation with the rural population (X4). F2 is the tourism dimension, which showed a strong positive correlation with the three driving factors including international tourism income (X12), passenger income (X13), and travel income (X14). In terms of contribution, F1 (81.17%) was much larger than F2 (13.53%). It can be concluded that the effect of F1 on land use change was more significant than F2.

**Table 4.** Rotated component matrix of the principal component analysis.

| Variables | Description | Component | |
|---|---|---|---|
| | | **F1-Socioeconomic** | **F2-Tourism** |
| X1 | Average annual temperature | 0.980 | −0.152 |
| X2 | Total annual precipitation | −0.865 | −0.338 |
| X3 | Gross population | 0.973 | −0.179 |
| X4 | Rural population | −0.975 | 0.221 |
| X5 | Urban population | 0.976 | −0.204 |
| X6 | Urbanization rate | 0.977 | −0.194 |
| X7 | Gross domestic product (GDP) | 0.968 | −0.248 |
| X8 | Agricultural value added | 0.962 | −0.185 |
| X9 | Industrial value added | 0.980 | −0.141 |
| X10 | Export of agricultural raw materials | 0.859 | 0.029 |
| X11 | Total fisheries production | −0.926 | −0.072 |
| X12 | International tourism income | 0.693 | 0.706 |
| X13 | Passenger income | 0.050 | 0.927 |
| X14 | Travel income | 0.722 | 0.669 |
| X15 | Main grain output | 0.983 | 0.053 |
| X16 | Natural rubber output | 0.920 | −0.178 |
| X17 | Main fruit output | 0.940 | 0.203 |
| Variance (%) | | 81.17% | 13.53% |
| Eigenvalues | | 13.63 | 2.30 |

The area of rainfed cropland, irrigated cropland, other cropland, forest, shrubland, and impervious surfaces, were 6 dependent variables respectively. Two principal components (F1, F2) were independent variables. We used multiple linear stepwise regression method to establish regression models respectively (Table 5). It followed that the rainfed cropland area showed a significant negative correlation with the socioeconomic (F1) and tourism (F2) factors. This indicated that the area of rainfed cropland decreased with socio-economic and tourism development. Forest areas showed a significant negative correlation with the socioeconomic (F1) factor. This indicated that the forest area decreased as the level of socioeconomic development in Thailand increased. Shrubland and impervious surfaces areas showed a significant positive correlation with the socioeconomic (F1) factor. This indicated that the area of shrubland and impervious surfaces increased in parallel with the socioeconomic development of Thailand. Shrubland area showed a significant negative correlation with tourism (F2). This indicated that the area of shrubland decreased with the development of tourism in Thailand. There was no plausible statistical relationship between irrigated cropland and other cropland and the principal components F1 and F2. In general, the land use/cover type area change in Thailand was mainly influenced by socioeconomic development.

**Table 5.** Relationships between land cover change and principal components in different types.

| Classes | Formula | $R^2$ |
|---|---|---|
| Rained cropland | $Y1 = 181,308.2$ *** $− 1750.5 ×$ F1 ** $− 964.6 ×$ F2 * | $R^2 = 0.99, p < 0.05$ |
| Forest | $Y2 = 180,854.2$ *** $− 1939.3 ×$ F1 * | $R^2 = 0.80, p < 0.05$ |
| Shrubland | $Y3 = 44,216.6$ *** $+ 2614.3 ×$ F1 ** $− 867.5 ×$ F2 * | $R^2 = 0.99, p < 0.01$ |
| Impervious | $Y4 = 14,386.3$ *** $+ 2284.9 ×$ F1 ** | $R^2 = 0.97, p < 0.01$ |

Note: * indicates passing the test of $p = 0.05$, ** indicates passing the test of $p = 0.01$, *** indicates passing the test of $p = 0.001$.

We used the twelve key driving factors obtained from the principal component analysis as the independent variables and the area of the six main land use/cover types as the dependent variables and applied the multiple linear stepwise regression method to build the model (Table 6). This showed that the rainfed cropland area showed a strong negative

correlation with the main grain output (X15) factor. Rice grown on irrigated land accounts for about 50% of Thailand's main grain output, so an increase in rice output will inevitably increase the pressure to shift out of rainfed cropland. Forest areas showed a significant negative correlation with the gross population (X3) factor. This indicated that an increase in the gross regional population would greatly enhance the pressure on deforestation. On the one hand, deforestation increased the area of food land, cash crop cropland, and economic forest land, and on the other hand, deforestation could also promote log exports and other wood-product-processing industries. The shrubland area showed a strong positive correlation with the industrial value added (X9) factor and a significant negative correlation with the passenger income (X13) factor. This showed that, on the one hand, when industrial development led to deforestation and the abandonment of arable land, these lands were transferred to becoming scrubland, and on the other hand, the development of international tourism usually led to the construction of infrastructure such as railroads, roads, and airports, and these construction activities also led to a reduction in shrubland area. The area of impervious surfaces showed a significant positive correlation with the urban population (X5) factor. This indicated that the urbanization of the population could lead to the urbanization of land, resulting in the expansion of impervious surfaces. As for irrigated cropland and other cropland, they did not have a plausible statistical relationship with the 12 driving factors currently identified.

**Table 6.** Multivariate linear stepwise regression results of the area of land use types and driving factors.

| Classes | Formula | $R^2$ |
|---|---|---|
| Rained cropland | $Y1 = 196{,}833.1\ ^{***} - 2.7 \times X15\ ^{**}$ | $R^2 = 0.95, p < 0.01$ |
| Forest | $Y2 = 229{,}311.8\ ^{***} - 7.3 \times X3\ ^{*}$ | $R^2 = 0.83, p < 0.05$ |
| Shrubland | $Y3 = 43{,}137.7\ ^{***} + 4.4 \times X9\ ^{***} - 147.0 \times X13\ ^{***}$ | $R^2 = 0.99, p < 0.001$ |
| Impervious | $Y4 = 4189.4\ ^{**} + 3.6 \times X5\ ^{***}$ | $R^2 = 0.99, p < 0.001$ |

Note: * indicates passing the test of $p = 0.05$, ** indicates passing the test of $p = 0.01$, *** indicates passing the test of $p = 0.001$.

## 4. Discussion

### 4.1. Land Use/Cover Change and Its Impacts

This study pointed out that from 2000 to 2020, the area of cropland, forest, and water bodies in Thailand showed an overall decrease, with the absolute area of forest decreasing the most; the remaining types of land area increased by varying degrees, with the absolute area of shrubland expanding the most. This result was consistent with the results of regional scale studies such as that in Tontisirin in eastern Thailand [32] and Tipaporn in the Thai Chi River basin [33] and the study by La et al. for the whole of Thailand during 1990–2019 [34]. All of their studies also showed a dramatic shrinkage of forest and cropland and an increased expansion of shrubland and impervious surfaces in Thailand. Thailand is a tropical and subtropical developing country and is largely consistent with similar developing countries in the world in terms of land use/cover type change. Studies by Hu et al. [35], Zhang et al. [36], and Niu et al. [37] demonstrated that the expansion of impervious surfaces and the shrinkage of cropland and forest land have also been observed in Vietnam, Laos, and Cambodia in the Indo-China Peninsula. For this reason, we propose that while the national economy is developing, land must be managed and planned reasonably; in particular, comprehensive and powerful measures must be taken to protect the country's forest resources.

Most of Thailand is covered by cropland and forest, and these two land types account for about 85% of the country's land area. Cropland and forest are key to sustainable economic and social development and regional ecological conservation in Thailand. This study pointed out that from 2000 to 2020, the areas of cropland and forest transformed into other land types were $2.45 \times 10^4$ km$^2$ and $2.25 \times 10^4$ km$^2$, respectively. Of these, cropland was mainly transferred out to forest and impervious surfaces, and forest was mainly transferred out to shrubland and cropland. This showed that the phenomena of returning

cropland to forest and logging areas to cropland existed at the same time, and there were also phenomena of destroying cropland for expansion and forest degradation. In 1999, the International Symposium on Participatory Land Use Planning and Land Allocation held in Bangkok pointed out that due to the lack of relevant land use policies and laws, many regions had problems such as unreasonable mechanisms for forest land allocation [38]. In addition, the excessive application of pesticides in cropland was an important issue affecting the sustainable development of Thai agriculture and the competitiveness of the country's exported agricultural products [39]. As a result, Thailand has combined forest conservation with tourism by establishing various types and levels of national parks, forest parks, botanical gardens, and tree gardens [40] as well as through local cultural and religious contexts in the form of "monks for forest conservation" and "trees for monks" [41] to protect Thailand's forest ecosystems. Nevertheless, it is still necessary to improve the laws and regulations so that the government, farmers, and other diversified subjects are involved, forming a market-oriented mechanism that benefits multiple parties.

*4.2. Driving Mechanism*

The analysis of the driving mechanisms of land use/cover change has been an issue of interest to scholars. However, the process of a country's development involves the influence of many factors, including natural and human ones. Therefore, the driving factors established are extremely complex and diverse. It was very difficult for us to find a direct and precise cause-and-effect relationship between them. We first conducted a principal component analysis and then used a multivariate stepwise linear regression to establish the relationship between land use change and climate change, and regional economic and social development factor changes. This approach simplified the selection process of the driving factors on the one hand, and at the same time allowed for an explicit and clear demonstration of the relationship between the driving factors and the driving results on the other hand.

Research on the driving mechanism of global land use change has shown that [42] 60% of all land changes were related to direct human activities and 40% were related to indirect drivers such as climate change. It has also been shown that the driving forces of LUCC in the "Golden Quadrangle" of China, Myanmar, Thailand, and the Lao People's Democratic Republic were population growth, socio-economic drivers, road construction, and slash-and-burn land use [43]. The main driving factors for land use/cover change in northeast Thailand were the extension and intensification of social agriculture [13]. Changes in LULC in Vietnam were mainly influenced by economic development and human activities, especially changes in the GDP, population, and urbanization rate [44]. We also corroborated the above findings. Our study pointed out that at the national scale, the land use/cover change in Thailand was mainly influenced by socio-economic factors, tourism development, etc., and it was less influenced by climate. Our study provides a deepening and refinement of the study of global LUCC and its driving mechanisms as well as a typical regional case study in the study of global LUCC.

Thailand joined the WTO in 1995. Since then, Thailand has gradually become more connected with the rest of the world. Thailand exports agricultural products, has a strong international tourism industry, and is also an important global producer and exporter of automobiles. Thailand's economy is growing rapidly, and the openness and tolerance of its society are increasing. All these factors have important implications for the land use/cover change in Thailand. Particularly, on 1 January 2022, Thailand signed the Regional Comprehensive Economic Partnership Agreement with China and other countries, which will further expand the scale and depth of the economic and population exchanges between Thailand and neighboring countries such as China, Cambodia, Laos, and Vietnam. Therefore, while carrying out international cooperation and developing tourism in the future, Thailand should also pay attention to its negative impact on ecological land and do a good job in protecting the high-quality ecosystems of virgin forests and border areas so as to achieve sustainable development.

*4.3. Uncertainty*

The authors of the GLC_FCS30 dataset experimentally demonstrated that the overall accuracy of the dataset was 82.5%, with 94% for forest, 88% for cropland, 56.8% for shrubland, 67.3% for grassland, 79.3% for impervious surfaces, and 83.8% for water bodies [18]. Florence et al. showed that almost all land cover datasets with cropland patches include a large number of pastures and other vegetation types (e.g., Paraguay and the Brazilian Amazon) [45]. Wang et al. also showed that the description of land composition was consistent across datasets, but the accuracy varied, and the accuracy and usability of the LULC dataset needed further improvement [46]. Pérez-Hoyos's study pointed out that different land use/cover datasets were different in terms of collection time, monitoring methods, and classification methods, which in turn led to differences in the classification results of different datasets [47]. The area of cropland and forest in Thailand accounts for 53.77% and 32.15% of the national land area, respectively; therefore, the accuracy of the base data, such as the mapping of cropland and forest, will likely have had an impact on the results of this paper.

The precipitation in Thailand during the rainy season (June to mid-October) accounts for 80% of the year, often causing floods and causing the rivers of the Chao Phraya River, Mon River, and Mekong River to rise sharply, causing disasters such as urban waterlogging, etc., while in the other two seasons, rainfall is lower, and drought even occurs [48]. Therefore, there was great uncertainty in the natural conditions for the study of changes in water bodies and wetlands. The area of water bodies is affected by the time phase of remote sensing images, and the determination of water levels in different periods (flood level, low water level, flat water level, etc.) has an important impact on the determination of water bodies and wetland areas. So, we did not analyze the sources and destinations of water bodies and wetlands, nor did we study their driving mechanisms.

In addition, there were limitations in the analysis methods of principal component analysis and multiple linear stepwise regression used in this paper. The current study only established a linear relationship between the values of the driving index and the values of the LULC area at the national scale but did not spatially correspond to the above-mentioned driving factors, as LULC changes one by one. At the same time, the existing methods are unable to incorporate human factors such as culture, religion, national policies, local planning, and people's intentions. These are important research directions that need to be deepened in the study of LUCC driving mechanism modeling, driving mechanism quantification, and driving mechanism spatialization [49,50].

## 5. Conclusions

Based on the authoritative GLC_FCS30 dataset and official socioeconomic statistics, we analyzed the spatial and temporal characteristics of land use/cover change in Thailand from 2000 to 2020, summarized the driving mechanisms of socioeconomic development on LULC change, and made corresponding recommendations for sustainable development and ecological conservation in Thailand. This paper was a typical regional case study in global sustainable development research on the process and driving mechanisms of LUCC in Thailand at the national scale and covering the longest time series.

This paper pointed out that the land use/cover types in Thailand were mainly cropland and forest, with cropland occupying more than 53% of the national land area. From 2000–2020, the rainfed cropland and forest areas shrank significantly, and impervious surfaces expanded rapidly. The LUCC changes in Thailand were mainly influenced by socioeconomic development and less by climate change; gross population, main grain output, industrial value added, passenger income, and urban population were the key driving factors of LUCC in Thailand.

In this paper, we established the spatio-temporal evolution analysis method of LUCC based on the GLC-FCS30 dataset and the driving mechanism analysis method based on a principal component analysis and a multiple linear stepwise regression analysis. These methods can be borrowed and applied to similar LULC change studies in other countries or

regions. Although these methods have some practicality, it is still worthwhile to carry out more in-depth research on the quantification of humanistic factors and the spatialization of driving relationships.

**Author Contributions:** Writing—original draft, Y.W.; software, Y.W. and X.N.; data curation, Y.W. and X.N.; conceptualization, Y.H.; methodology, Y.H. and Y.W.; writing—review and editing, Y.H. and Y.W.; supervision, Y.H. and L.Z.; project administration, Y.H.; funding acquisition, Y.H.; formal analysis, H.Y. and L.Z.; investigation, L.Z. and H.Y.; visualization, X.N. and H.Y. All authors have read and agreed to the published version of the manuscript.

**Funding:** This research was supported by the National Natural Science Foundation of China (42130508), Strategic Priority Research Program of the Chinese Academy of Sciences (XDA20010202) and Network Security and Information Program of the Chinese Academy of Sciences (CAS-WX2021SF-0106).

**Data Availability Statement:** Not applicable.

**Acknowledgments:** The authors would like to express their sincere thanks to the anonymous reviewers because the comments and suggestions were of great help in improving the quality of this paper.

**Conflicts of Interest:** The authors declare no conflict of interest.

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
