# Peer review of "Land Use/Cover Change and Its Driving Mechanism in Thailand from 2000 to 2020"

_land, doi:10.3390/land11122253_

Round 1

Reviewer 1 Report

Dear Authors,

I have made some comments and suggestions mentioned below on your manuscript:

1. The period of time series analysis is not clearly mentioned (2000 & 2020? or 2000-2010-2020?). 

Why 5 years time interval was not considered? though the land use at 5 years time interval is available since 1985 and the other socio-economic data is available at annual time scale?

2. How the rainfall and temperature trend was estimated? 

3. More clarity is needed for the data used and methodology : 

- Line 177: Why bare area were merged in shrublands? How Table 1 and Table 2 are related? How the GLC_FCS classification system was compared with the LULC classification system of Thailand?

- Which global product was used for which meteorological variable? either GLDAS or PERSIANN-CDR? which is used for temperature and which one for precipitation? Why GLDAS or PERSIANN was considered though there are several global products available for precipitation and temperature. Please justify your selection.

- Line 184: Why calculated day by day? as the monthly global data is already available.

- Mention the variables from X3-X17 in text and justify the reason for their selection in this study 

- Mention clearly in text the different driving factors considered in this study. Also a justification for their selection

- Write in a bit more detail about the multiple linear regression and stepwise regression analysis 

- Please make it clear what is dependent variable and what are independent variables in text

4.  I don't see a comprehensive uncertainty analysis carried by you. However this a research question listed in your paper.

GLC_FCS30 dataset provides an average accuracy assessment of their product for different land use classes and not specific to your study area. 

Author Response

Thanks for your review. Your questions and suggestions are very helpful to improve the quality of the article. We attach great importance to this revision opportunity and have carefully checked and revised it. Please find your original comment (black) and our reply (blue) in the uploaded Word. All line numbers refer to the line numbers in the revised manuscript.

Reviewer 2 Report

This paper examined land use and cover change in Thailand using various indicators and methods.  The main conclusions were a summary of the changes that have occurred in Thailand over 20 years and an establishment of spatio-temporal evolution analyses that can be applied to other areas.  The strength of this paper is the calculated results for those interested in Thailand and the dynamic change analysis that was performed.  The main issue I have with the work is a lack of originality and contribution of the work that would make others interested in reading or citing this effort.

Below are more specific areas of concern I have with this paper:

Line 23: mutual conversion of rainfed cropland and forest is a confusing phrase.

Line 26: transfer out is a confusing phrase

Line 29-31: Not much was ever developed in this paper to address how the research can provide the basis and decision support for the future planning and management of land.

Line 78-80: I am not convinced by the background information that this study truly provides new ideas for LULCC dynamics research.  

Line 105: The problems of LULC research were not fully expanded upon and how this study will specifically address the need.  This section really needs to be enhanced so the reader is assured of how this paper contributes to our information base and provides something new.

Line 112-116: Why was Thailand chosen for this study?  Why and how is it appropriate for this work?  Why only 20 years?  How does this compare to other LULCC studies that have been done regarding the amount of time?  What are some of the main mechanisms or uncertaintees that other studies have found to note?

Line 157: Why is topography shown first in this map and not the previous or most current land cover?  The labels are difficult to see too.

Line 170-180: The US uses USGS LULC Anderson Level I and Level II classification for classes.  Why does this Thailand approach not follow that logical approach to nesting classes?

Line 200: Based on what stats?  I was expecting a hypothesis test and equations to test that Ho?

Line 262-273: These calculations are something to be in a report and not really a research article that is supposed to be testing or informing us of something new.  I would bet that this work has been done by someone, somewhere already.

Line 300-308: Why not show the reader this information in a series of map figures?  There are some layers of discussion that are introduced here that are really not expanded upon at all.

Line 309-350: I think this is the best part of this paper and feel like with some better editing, this should be the main contribution of this work. Asking additional why questions and posing some better tests to further examine what this means from a regional to local perspective could be interesting.  

LIne 358: This figure 4 is a mess. There has to be a better way to represent the cover class changes over these years.

Line 384 on: Relating temperature to land cover change and population should be flushed out more and made more of a focus of this study.

LIne 400-407: I am not sure why this GDP metric was brought into the discussion.  There was not much of an introduction to why it is in this study and what result was concluded from it.

LIne 431-492: Having some correlation and causation hypothesis test could help this section have more of a profound impact in this study and provided a better template for others to consider to use.  There is potential in this section to really expand this as the main focus of the paper and build around it as an area of emphasis.

Line 515: I don't feel this statement can be logically stated without more evidence provided for the reader.

Line 567: Uncertainty is something that should be put in context for this study regarding the pixel cell size, accuracy of classes, number of grid cell alignment across grids, referencing, propagation of errors, etc.

Line 599-: Way too much summary in this conclusions section and not enough of the noted contribution of this work for others to follow and use.  It was noted that the methods can be borrowed and applied to similar LULCC studies but it did not provide a convincing and template to deal with different land cover years, size extent, resolution, accuracy, urban vs natural landscape, etc to really make this statement that the work is that transferable to other studies.

Author Response

Thank you for your suggestions and comments. You have reviewed the words and sentences in the article very carefully, which is very helpful to improve the language expression of the article. We have incorporated and addressed these comments in the revised manuscript. Please find your original comment (black) and our reply (blue) in the uploaded Word. All line numbers refer to the line numbers in the revised manuscript.

Reviewer 3 Report

This manuscript systematically provided the temporal and spatial patterns and their driving mechanism of LUCC in Thailand from 2000 to 2020, an overall picture of the mutual transformation of rain-fed cropland, irrigated cropland, forest shrub-land, and wetlands. It deserves publication in Land after some revision.

But there are some problems.

(1) The manuscript is too long, it must be greatly shortened.

 (2) Some sentences are difficult to understand or even wrong in grammar, e.g., in Abstract, “The LUC with the largest transfer out area is forest and the LUC with the largest transfer into the area is shrub-land”, “ From 2000 to 2020, the process in Thailand has been mainly influenced by socio-economic and tourism. what process?

 (3) The section “Introduction” is too long. After more than 30 years of study, LUCC is well understood worldwide, so it is not necessary to describe it in such a detail.  

(4) Fig.1 needs improvement. The size of the abbreviations is too large. And what is the unit for DEM? Please pay great attention to the relatively independence, information integrity and readability of figures in scientific papers.

(5) I find that analysis of LUCC driving forces is rather weak in almost all LUCC papers. This is also the case for this manuscript. It is very difficult to quantitatively relate LUCC with several factors. This greatly undermine the study of LUCC, especially on small scales.  

 (6) Frankly speaking, Chinese scientists have spent too much time on the study of LUCC in recent 30 years. Theories and methods associated with LUCC have not been well advanced. I really hope to see the appearance of new theories and methods in LUCC study. 

Author Response

Thank you for your valuable comments. These comments are very helpful to improve the quality of the article. We have processed your comments and suggestions. Please find your original comment (black) and our reply (blue) in the uploaded Word. All line numbers refer to the line numbers in the revised manuscript.

Round 2

Reviewer 1 Report

Dear Authors, thanks for considering my comments and suggestions. I feel satisfied with the revised version. However, I have still some minor comments/suggestions mentioned below:

1. Please write an overall objective of this research before the research questions. Line 110-114, you mentioned the proposed methdology which should come later in the methodology section. 

2. If you have not conducted uncertainity analysis, then why did you listed it as a research question? As you talked about the global accuracy of LULC reported by the developers of LULC map, it could be mentioned in the data description section only. 

Author Response

Thank you for your review. The questions and suggestions you raised last time are very helpful to improve the quality of the article. We have carefully examined the two questions you raised this time. Please find your original comment (black) and our reply (blue) at. All line numbers refer to those in the revised version.

Reviewer 2 Report

Issues were addressed.

Author Response

Thank you for your review of our manuscript. Your suggestions and comments last time are of great help to the quality of our articles. We have benefited a lot, and we will pay attention to related issues in the future manuscript writing process.